# Enabling Control Co-Design of the Next Generation of Wind Power Plants

Andrew P. J. Stanley, Christopher J. Bay, and Paul Fleming

National Wind Technology Center, National Renewable Energy Laboratory, Golden, CO, 80401, USA

**Correspondence:** Christopher J. Bay (Christopher.Bay@nrel.gov)

**Abstract.** Layout design and wake steering through wind plant control are important and complex components in the design and operation of modern wind power plants. They are currently optimized separately, but with more and more computational and experimental studies demonstrating the gains possible through wake steering, there is a growing need from industry and regulating bodies to combine the layout and control optimization in a co-design process. However, combining these two op-
timization problems is currently infeasible due to the excessive number of design variables and large solution space. In this article, we present a method that enables the coupled optimization of wind power plant layout and wake steering with no additional computational expense than a traditional layout optimization. We developed a geometric relationship between wind turbines to find an approximate optimal yaw angle, bypassing the need for either a nested or coupled wind plant control optimization. It also provides a significant and immediate improvement to wind power plant design by enabling the co-design of
turbine layout and yaw control for wake steering. A small co-designed plant shown in this article produces 0.8% more energy than its sequentially designed counterpart. This additional energy production comes with no additional infrastructure, turbine hardware, or control software; it is simply the outcome of optimizing the turbine layout and yaw control together, resulting in millions of dollars of additional revenue for the wind power plants of the future.

## 1   Introduction

Optimizing the layout of wind turbines within a wind power plant is a highly complex problem wherein the wind plant developer must weigh numerous competing goals and constraints against each other. One common objective is maximizing the expected energy production of the plant while minimizing the cost to build. The problem also includes constraints on layouts, which could include specified boundaries, wind turbine spacing requirements, grids or other layout regularity, setback from shipping lanes or structures, and seafloor or terrain-based constraints.

Predicting the impact of wind turbine wakes on total wind plant production plays a key role in wind plant design. Within a wind plant, wind turbines interact with each other through the wakes that they produce while extracting energy from the passing flow (Sanderse et al., 2011). These wakes have reduced wind speed, which limits the energy that is available to downstream turbines in the plant; additionally, these wakes have higher turbulence than the ambient flow, which increases loads and is detrimental for structural reliability. Negative wake impacts can be mitigated in a wind plant's design phase as well as during
plant operation.

The primary way to minimize wake interactions during the plant design stage is through wind turbine layout optimization, often referred to as micrositing, which is an important step for both offshore and land-based wind power plants (Hou et al., 2019; Balasubramanian et al., 2020). Through layout optimization, wake interactions can be minimized for the wind resource, atmospheric conditions, turbine design, and constraints unique to a specific site. As mentioned earlier, wind plant layout optimization is notoriously challenging because of the large number of interacting variables as well as the complexity of the required models and design space.

During operation, wind power plant control can be used to reduce wake interactions. One plant-level control strategy is yaw misalignment for wake steering. A wind turbine whose yaw angle is misaligned with the incoming wind will produce a wake that is deflected compared to an unyawed turbine. This phenomenon can be exploited to intentionally steer wakes away from downstream turbines in the wind plant. Although a wind turbine with some yaw misalignment to the incoming wind will suffer reduced power production and increased loading, wake steering can result in a net improvement for the entire plant. This improvement has been demonstrated with several different fidelities of wind plant simulations (Jiménez et al., 2010; Gebraad et al., 2017; Martínez-Tossas et al., 2021) as well as with wind tunnel experiments (Campagnolo et al., 2020). Because of these promising simulations and experiments, wake steering is being adopted more frequently at existing sites. For example, there have now been several demonstrations of wake steering implemented at commercial wind power plants (Fleming et al., 2017, 2019, 2020; Simley et al., 2021; Howland et al., 2022). There have also been several announcements of commercial implementations of wind plant control, either provided by the wind turbine original equipment manufacturers or consultants.

An enormous opportunity for improved wind plant performance presents itself by simultaneously optimizing wind plant layout and turbine yaw angles. Generally, this process is called control co-design, which means to account for aspects of system control throughout the entire design process (Garcia-Sanz, 2019). Specifically, control co-design can be leveraged to maximize the capture of spatially varying wind resources, such as offshore sites with wind speed correlated to the distance from shore, or complex terrain where higher wind speeds can exist on higher elevation topologies. Control co-design would allow for operational wake loss mitigation to be considered during the layout optimization. Control co-design can also make better use of the available space in lease areas where lease fees are significant, or reduce installation costs by condensing wind turbines into shallower offshore regions. Coupled with other design parameters and constraints, the possible benefits of control co-design are numerous.

Currently, the possibility of control co-design is severely limited in wind plants by the large number of design variables required to fully couple wind plant layout and yaw control optimization. In its most basic form, optimizing wind plant layout and yaw angles requires two design variables for every wind turbine (one for both the $x$ and $y$ coordinate), and one design variable per turbine *per wind speed and wind direction combination* for the yaw angles. This relationship means that the computational expense required to run the fully coupled optimization scales very poorly as the number of turbines increases, a challenge often called the "curse of dimensionality." For an average-sized wind plant ($\sim$ tens of turbines), the fully-coupled problem can easily reach thousands or tens of thousands of coupled design variables. Figure 1 shows the wall time required to run a fully coupled layout and yaw control optimization versus the number of wind turbines. These optimizations were run with the gradient-based Sparse Nonlinear OPTimizer (SNOPT) (Gill et al., 2005, 2018) within the pyOptSparse optimization framework (Wu et al.,

2020) with finite-difference gradients. The objective was maximizing plant energy production modeled with FLOw Redirection and Induction in Steady State (FLORIS) (National Renewable Energy Laboratory, 2022), a controls-focused wind plant simulation software incorporating steady-state engineering wake models with wake deflection modeling capabilities. These optimizations were run on a single core with no parallelization on the high performance computer at the National Renewable

Energy Laboratory. The CPU used was a Dual Intel Xeon Gold Skylake 6154 (3.0 GHz, 18-core) processor. This figure highlights two characteristics of this problem. First, the time to optimize scales non-linearly with the number of turbines. Second, even with the small wind plants optimized in the creation of this figure, the wall time for the fully coupled problem is far too long for most applications. While the absolute value of this metric could be reduced through advanced computing capabilities and finely tuned optimizer settings, the principle remains that the fully coupled optimization problem is computationally

expense, especially for large wind plants.

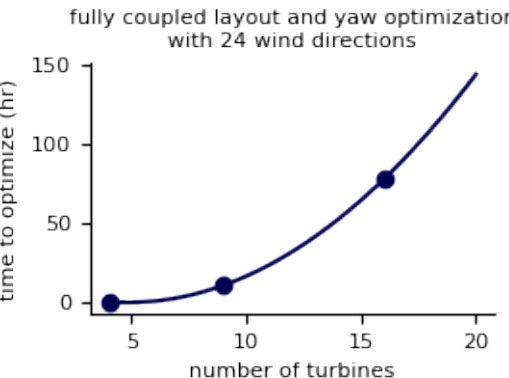

**Figure 1.** The time to solve the coupled wind turbine layout and yaw control optimization problem as a function of the number of wind turbines. These optimizations were performed with the gradient-based Sparse Nonlinear OPTimizer (SNOPT), using 24 wind direction bins and one wind speed bin for each wind direction. The three points represent the fully couple optimizations that we ran (with 4, 9, and 16 wind turbines), while the line shows an exponential fit to the three points. The 4 turbine optimization fully converged, but the 9 and 16 turbine optimizations finished with numerical difficulties due to the complexity of the optimization problem.

In practice, this description of the scaling issue of the fully coupled optimization understates the problem, as real-world wind plant design must include many design constraints, some of which were already mentioned. Additionally, in practice, wind plant layout optimization is often performed over an already-existing developer-specific software tool set, making co-design of the layout and yaw control even more prohibitive because practical implementation requires a control optimization

nested within the existing plant optimization.

In this article, we present a novel method to determine yaw angles for wake steering that enables layout and yaw control co-design with no increase in computational expense compared to the layout only optimization. This new method is to define wind turbine yaw angles deterministically from the layout of a wind plant, meaning that yaw control can be considered during layout optimization with no additional design variables. This coupled, efficient optimization is frequently being requested by

industry and is extremely relevant for the next generation of offshore and land-based wind power plants that are looking to

maximize the wind generation in a limited space. Expensive and limited lease areas (Friedman, 2022), increasingly strict siting regulations (Mai et al., 2021), and improved technology enabling larger turbines (Enevoldsen and Xydis, 2019) will drive the wind plants of the future to have turbines packed close together relative to the rotor diameter, a situation where wake steering is particularly effective. A small example plant with 16 turbines presented in this article produced 0.8% more energy when the layout and yaw angles were optimized together than when the layout and yaw angles were optimized sequentially. To put this in perspective, 1 MW of wind capacity generates annual revenue on the order of $100,000. Therefore, a 0.8% increase in performance equates to an additional $800 per MW, or $800,000 per GW each year.

## 2  Geometric Yaw Relationship

When optimizing the yaw offset angles in a wind plant, there are many different combinations of turbine yaw angles that result in almost identical plant performance. We determined that a sufficiently optimal yaw angle for any individual wind turbine can be calculated as a function of the streamwise and cross-stream distance to its nearest downstream waked turbine, as shown in Figures 2A and 2B. Figure 2A shows a group of five turbines with the wind coming from the left. To determine the yaw angle of the yellow turbine using our geometric yaw relationship, it is necessary to calculate the streamwise distance (dx) and the cross-stream distance (dy) to the nearest waked turbine shown in purple. The black circles represent the other wind turbines in this cluster. Notice that there are two turbines closer to the yellow turbine, but these are not waked and therefore do not affect the yaw angle of the yellow turbine. To determine if a turbine was waked, we assumed a wake radius of $r_{\text{wake}} = 0.1x + r_{\text{turbine}}$, where $r_{\text{wake}}$ is the radius of the wake, $x$ is the streamwise distance downstream of the waking turbine, and $r_{\text{turbine}}$ is the radius of the waking turbine (Jensen, 1983). As is demonstrated in the following paragraphs, this definition of the wake radius is sufficiently wide to explain the development our geometric yaw relationship. Figure 2B shows the same group of five turbines, but with the wind coming from the upper left corner. For this wind direction, the nearest waked turbine is different than in Figure 2A.

To understand the relationship between optimal wind turbine yaw angles and their position relative to the nearest downstream waked turbine, we optimized the yaw for many different wind plants, including randomly generated layouts with different numbers of turbines, average turbine spacings, and wind speeds, as well as regular grid layouts with different numbers of rows and columns, turbine spacings, grid rotations, and wind speeds. As with the fully coupled optimizations discussed previously, these yaw angles were optimized with the gradient-based Sparse Nonlinear OPTimizer (SNOPT) within the pyOptSparse optimization framework (Wu et al., 2020). The objective was maximizing plant power modeled with FLORIS. For these yaw optimizations, we again used finite-difference gradients, bounds between -30 and 30 degrees for turbine yaw angles, and default convergence settings. For additional information regarding these optimizations, please see the run scripts in the code referenced at the end of this paper. The result was over 100,000 optimized wind turbine yaw angles.

Figure 2C shows the yaw angles we optimized as a function of position of the yawed turbine relative to its nearest waked downstream turbine, normalized by the turbine rotor diameter. A single point in this figure represents the yaw angle of a single turbine, represented by the color, as a function of the distance to the nearest downstream waked turbine, which is indicated by

the point's position on the plot. A clear pattern emerges from Figure 2C. There is a divide between positive and negative yaw angles depending on whether the cross-stream distance to the nearest waked wind turbine is positive or negative. Additionally, we can see that the turbine is only yawed if the cross-stream distance to the nearest downstream waked turbine is around one rotor diameter or less. Outside of that range, the upstream turbine has an optimized yaw angle near zero. As can be seen by the handful of orange and purple points beyond the one-rotor-diameter threshold, there are a few exceptions to this observation. However, in rule seems to apply most of the time. Observing this pattern, we created the geometric yaw relationship shown in 2D, which can be used to instantly determine a near-optimal yaw angle for a wind turbine as a function of its location relative to the turbines around it. The specific relationship is a one-dimensional gradient starting at the upstream turbine with a value of 30 degrees, and linearly decreasing to 0 degrees at 25 rotor diameters downstream. The sign of the yaw angle is determined by the lateral placement of the downstream turbine, as shown in Figure 2D.

As explained, the relationship shown in Figure 2D was manually constructed simply by observing the pattern apparent from the continuously optimized yaw angles shown in Figure 2C. This is an intentionally simple approach to define a relationship between relative turbine locations and quasi-optimal turbine yaw angles. This paper is intended to demonstrate the concept that yaw angles can be implicitly defined from the turbine layout such that wake steering can be considered during layout optimization, not to claim that this specific geometric yaw relationship is the best relationship possible. We expect and hope that this concept will be expanded to include additional dimensions and more sophisticated methods to determine the turbine yaw angles. Additionally we expect future work to expand upon this geometric yaw relationship include other operational scenarios, such as other forms of control, or scenarios where a wind plant operates with advanced control strategies only a portion of the time. The important message is that even with the simple relationship that we introduce, significant gains are already achieved compared to optimizing layout and yaw control sequentially, as explained in the following section.

Please also note that in this paper, we optimized wind plants solely for the benefit of power production objectives and have ignored any impacts layout optimization and active wake steering may have on turbine loads. There is increasing interest in including loading impacts into the optimization (Navalkar et al., 2023). We expect that a similar geometric yaw relationship to the one we present here could be used in such an optimization that also considers turbine loading. Figure 2C shows the data points used to manually create the geometric yaw relationship used in this paper. The yaw angles shown in this plot were all optimized to maximize power. In order to include turbine loads, or any other consideration, this data used to intuit or train the geometric relationship would just need to be optimized for the desired objective instead of maximizing power as we have done.

## 3 Results

In this section, we present two examples of implementations of geometric yaw in the wind plant layout optimization problem. Like many other scenarios, the scenarios presented in this article perform better and look significantly different when the layout and yaw control are optimized together. Additionally, the use of our geometric yaw relationship during turbine layout optimization allowed these examples to be run on a laptop on a single processor, which is infeasible with existing methods.

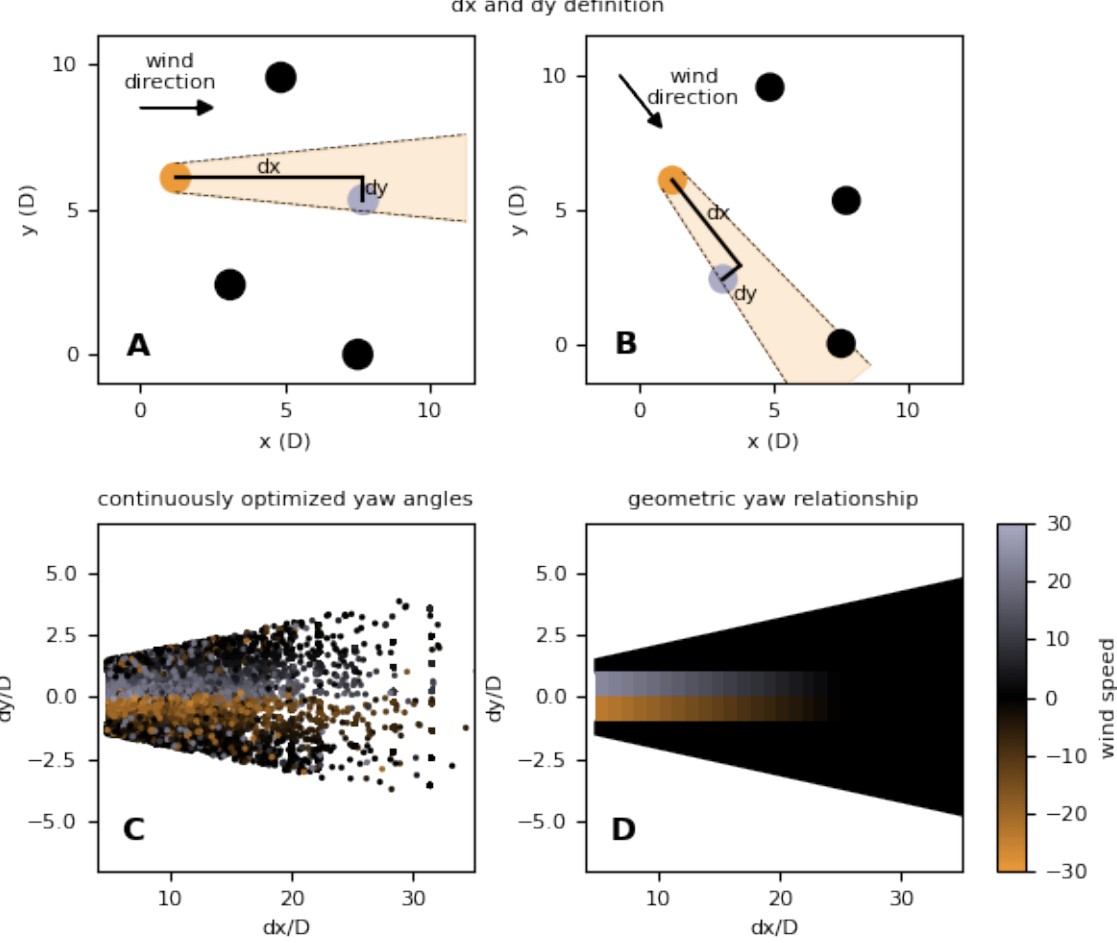

**Figure 2.** A description of the geometric yaw relationship presented in this article. Figures 2A and 2B show how the streamwise (dx) and cross-stream (dy) distances to the nearest downstream waked wind turbine are defined for two different wind directions. Figure 2C shows the optimized yaw angles for over 100,000 individual turbines optimized continuously in a variety of wind plants with different numbers of turbines, layouts, turbine spacing, and wind speeds. These yaw angles are shown as a function of the streamwise and cross-stream distance to the nearest downstream waked turbine of the yawed turbine. Figure 2D shows the geometric yaw relationship that we defined by observing the pattern that emerges in Figure 2C, which can be used to immediately determine a near-optimal yaw angle of any turbine in a wind power plant.

For each example in this section we used the SciPy (Virtanen et al., 2020) SLSQP gradient-based optimizer within the py-OptSparse optimization framework with finite-difference gradients and default optimizer settings. All optimizations converged within the default convergence tolerance of 1E-6. Please refer to our code referenced at the end of this paper to see the our exact implementation and objective functions, including scaling. To model the plant performance, we used the Gauss-Curl-Hybrid model in FLORIS version 3.1.

For both of the examples, we compare two different wind plant layouts and how they perform. The first is a layout that was optimized assuming no yaw control. After the layout was optimized, the wind turbine locations were fixed and the yaw angles were optimized continuously to determine the final plant performance. The second layout was optimized using our geometric yaw relationship to define the yaw angles during the layout optimization. After the layout was optimized and fixed, one final continuous yaw optimization was performed to determine the final yaw angles and plant performance. With the geometric yaw relationship shown in Figure 2D, the purpose is to sufficiently account for yaw control during the layout optimization to affect the optimal turbine locations. Continuously optimizing the yaw angles for wake steering outperforms those predicted by the geometric yaw model, so at least with this specific relationship the final continuous yaw optimization was necessary after the layout was fixed. Perhaps an improved geometric yaw relationship could remove the necessity of this last yaw angle optimization.

### 3.1 One-Dimensional Plant

The first example we present is a simple one-dimensional wind plant. Although this problem would not occur in the real world, it is valuable to demonstrate the power of coupled layout and yaw optimization. In this example, 16 turbines were arranged in a straight line with constant wind in line with the row of turbines. The objective was to maximize the power density of the array, which was defined as the total power divided by the length of the row of turbines. The spacing between each adjacent turbines was assumed to equal, meaning there was one design variable in the optimization. To avoid convergence to local minima and lend confidence that our solution was close to the global optimum, we repeated the layout optimization 50 times with a randomly initialized starting spacing between 3 and 8 rotor diameters. We performed these 50 optimizations both for the layout only optimization and layout optimization using the geometric yaw relationship. Figure 3 shows the results of this optimization. Figures 3A.1 and 3A.2 show the layout that was optimized assuming no yaw control, whereas Figures 3B.1 and 3B.2 show the layout that was optimized using the geometric yaw relationship. Figures 3A.1 and 3B.1 show the plant and performance for each optimized layout without yaw control. Figures 3A.2 and 3B.2 show these same layouts, but with the final optimized yaw angles.

This example clearly demonstrates two important principles. First, the layout optimized without wake steering, which has the wind turbines spaced relatively far apart (Figures 3A.1 and 3A.2) is very different than the layout optimized with wake steering, which has the turbines much closer together (Figures 3B.1 and 3B.2). It is evident that the difference between the layouts is significant, indicating that including wake steering during the layout optimization can lead to a different solution. Second, the layout optimized with geometric yaw outperforms the layout optimized without geometric yaw by 6.5% when the plant is operated with wake steering (Figure 3B.2 compared to Figure 3A.2). However, when the plant is operated without

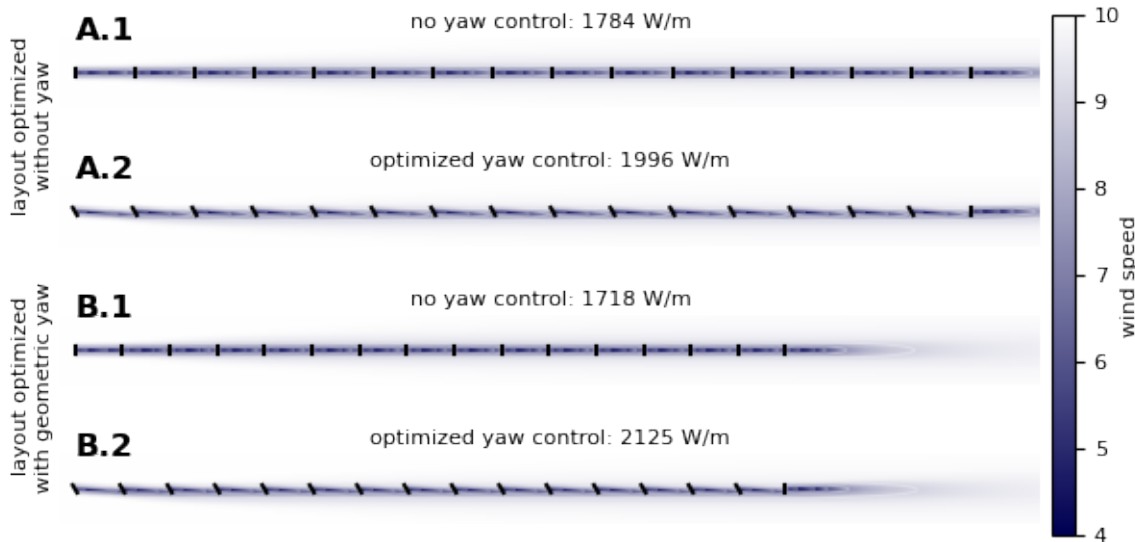

**Figure 3.** A simple one-dimensional optimization of a 16-turbine wind power plant comparing the performance of the layout optimized without yaw control and the layout optimized with geometric yaw. Figures 3A.1 and 3A.2 show the layout that was optimized assuming no yaw control. Figures 3B.1 and 3B.2 show the layout that was optimized using the geometric yaw relationship. Figures 3A.1 and 3B.1 show the plant and performance for each optimized layout without yaw control. Figures 3A.2 and 3B.2 show these same layouts, but with the final optimized yaw angles.

yaw control, the layout optimized without geometric yaw outperforms the one optimized with geometric yaw by 3.7% (Figure 3A.1 compared to Figure 3B.1). From this observation, we can conclude that the layout should be optimized with the yaw control scheme that will be used during plant operation. Plants that will be operated with wake steering will benefit greatly from optimizing the wind turbine layout with geometric yaw.

Figure 4 shows the computational expense to run each of the 50 randomly initialized optimizations with and without geometric yaw. As seen in this figure, all of these optimizations required comparable function calls and wall times, regardless of whether geometric yaw was used or not. In total, all 50 optimizations without geometric yaw converged in 937 function calls and 189 seconds, while those with geometric yaw converged in 949 function calls and 187 seconds, demonstrating that our geometric yaw method does not introduce additional computational expense.

### 3.2 Gaussian Hill Spatially Varying Inflow

The second example is a more realistic two-dimensional layout optimization with a full distribution of wind directions and spatially varying free-stream wind speeds across the domain. In this example, we optimized the layout of a wind plant with 16 turbines, with the objective to maximize the annual energy production of the plant. The turbines were constrained within a 2-by-2-km square, and had a minimum spacing constraint of two wind turbine rotor diameters. We used a bimodal wind rose

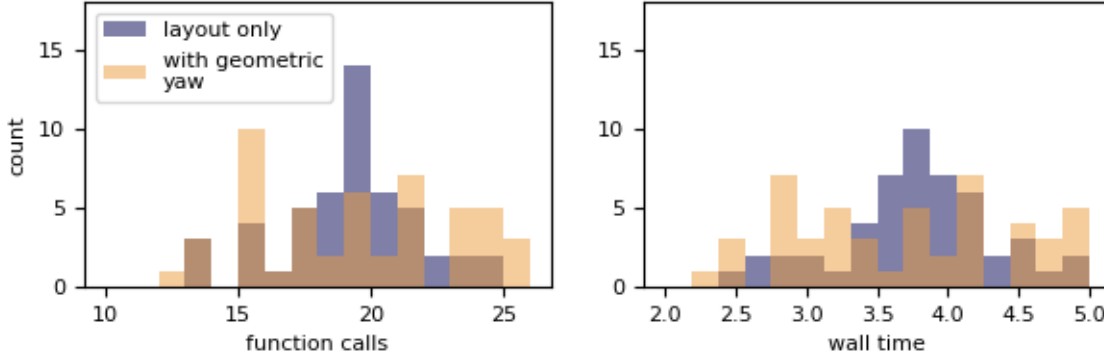

**Figure 4.** Histograms showing two computational expense metrics associated with running the one-dimensional layout optimizations shown in Figure 3. The subfigure on the left shows the number of function calls to convergence, while the subfigure on the right shows the wall time. The purple histograms represent the layout only optimizations, while the orange represent those with geometric yaw.

shown in Figure 6A divided into 72 discrete bins. From each wind direction we assumed a constant wind speed (indicated by the color bar in Figure 6A). In addition to the full wind rose, we assumed there was a spatially varying wind speed throughout the domain for each wind direction. This spatial wind speed variation was modeled by applying a Gaussian wind speed multiplier to the domain with a standard deviation of 600 m in each direction, which provided a maximum wind speed increase in the wind speed multiplier of 0.4 at the origin. This wind speed variation was meant to approximately simulate the spatial variation in wind speeds caused by a hill, including the speedup and wind shadow regions, so we also applied a penalty behind the hill to capture the wind shadow. For the penalty, we applied a second Gaussian distribution 400 m directly behind the origin in line with the wind direction. This negative Gaussian distribution again had a standard deviation of 600 m and provided a maximum decrease in the wind speed multiplier of 0.2. The interaction of these two Gaussian distributions is a maximum wind speed multiplier of about 1.25 and a minimum that is slightly less than 1.0. The resulting wind speed multiplier distribution for wind coming directly from the left is shown in Figure 6B (note that this figure only shows the speedup/slowdown for one direction; the location of the highest speedup and wind shadow change with the wind direction).

With the scenario fully defined, we optimized the plant layout both while assuming no yaw control and while using the geometric yaw relationship. Because of the large amount of local minima known to exist in the wind plant layout optimization problem, and because gradient-based optimizers are known to converge to local minima without full exploration of the design space, we repeated each optimization 50 times with randomly initialized design variables. This process was as follows:

1. Randomly initialize a starting turbine layout.

2. Perform the layout optimization twice from the starting layout in step 1, once with no yaw control and once with yaw control using the geometric yaw relationship.

3. Repeat steps 1 and 2 for a total of 50 different starting layouts.

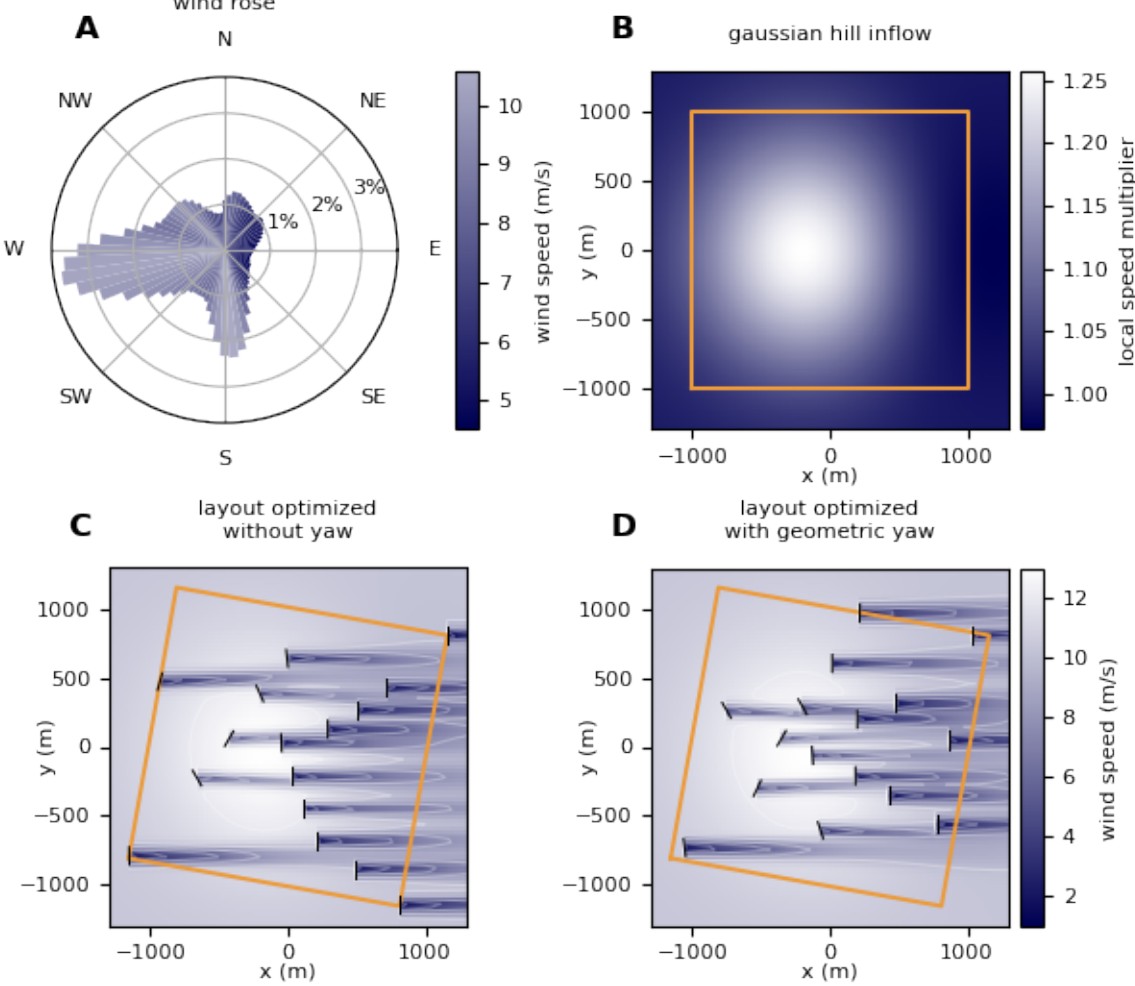

**Figure 5.** A wind turbine layout optimization of a 16-turbine wind power plant comparing the performance of the layout optimized without yaw control and with geometric yaw. In this figure, there is a spatially varying wind speed multiplier applied over the domain. Figure 6A shows the wind probability rose used in this optimization. The wind directions were divided into 72 5-degree bins, with directionally averaged wind speeds indicated by the color bar. Figure 6B shows the Gaussian wind speed multiplier applied to the domain for one wind direction (from the left). In our simulation, the wind shadow rotated behind the Gaussian peak for each wind direction. Figures 6C and 6D show the optimal wind turbine layouts, and optimal turbine yaw angles and flow fields for the dominant wind direction. Figure 6C shows the layout optimized without yaw, and Figure 6D shows the layout optimized with geometric yaw.

Optimizing the each of the starting layouts with and without yaw control removed the possibility that differences in the optimized layouts and performances were due to different starting conditions. The best-performing layout from each method was selected as the final plant layout, to which we performed one final yaw control optimization to evaluate the final plant performance. With existing methods, this coupled wind turbine layout and yaw angle optimization problem would have needed 1,184 fully coupled design variables, including two for each wind turbine to define the locations and one for each of the 72 wind directions to define the yaw angles. With our geometric yaw relationship we reduced that down to just 32 variables, the N×2 required for each turbine to define the layout, which allowed us to perform the optimization on our local machine with finite-difference gradients.

The layout optimized with geometric yaw produced **0.8% higher** AEP than the layout optimized without yaw. As previously discussed, because wind power plants are enormous investments, a performance gain around 0.8% can easily equate to hundreds of thousands or millions of dollars annually depending on the plant capacity. This particular performance improvement is even more impressive in that it does not require any additional components or technology, it simply involves placing the wind turbines in better locations that were not found before this geometric yaw relationship. The optimized turbine locations and the associated yaw angles and flow field for the dominant wind direction are shown in Figures 6C and 6D. Figure 6C shows the layout that was optimized without turbine yaw, and Figure 6 shows the layout that was optimized with the geometric yaw relationship. In these figures, the wind plant boundary is represented by the orange squares, which appear rotated because the dominant wind direction is from 260 degrees, and the wind in this image is coming from the left. Notice the extremely different layouts obtained with the two different optimization methods. Because the layout in Figure 6D was optimized with geometric yaw, the optimizer was able to place wind turbines closer together near the peak in the Gaussian wind speed multiplier, taking advantage of wake steering to reduce wake interactions between nearby turbines. On the other hand, the layout in Figure 6C was optimized without yaw. In this case, the optimizer did not take as much advantage of the wind speed multiplier and instead opted to spread turbines perpendicular to the dominant wind direction as displayed in the figure. This more complex example reiterates the conclusions found in the one-dimensional example—that optimizing the layout concurrently with wake steering leads to different optimal layouts and significant performance improvements.

Figure 6 shows the computational expense to run each of the 50 randomly initialized optimizations with and without geometric yaw for this two-dimensional optimization with spatially varying inflow. As with the simple one-dimensional example in the previous section, Figure 6 also demonstrates the similar computational expense in running the wind plant layout optimization with and without geometric yaw. The histograms for the layout only optimizations in the figure actually have longer tails to the right, meaning that for our setup to solve this layout optimization problem, using geometric yaw actually had lower computational expense. Although we do not expect a reduction in computation expense for other to be typical, these results along with those shown in Figure 3 together demonstrate that the geometric yaw relationship that we present enables couple wind plant layout and yaw control optimization with no increase in computational expense compared to the traditional layout only optimization problem.

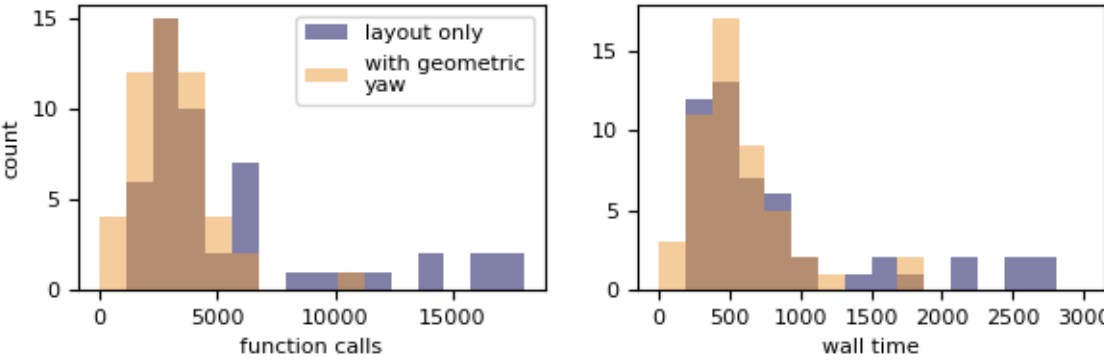

**Figure 6.** Histograms showing two computational expense metrics associated with running the layout optimizations with the Gaussian wind speed multiplier shown in Figure 3. The subfigure on the left shows the number of function calls to convergence, while the subfigure on the right shows the wall time. The purple histograms represent the layout only optimizations, while the orange represent those with geometric yaw.

## 3.3 Potential Impact at Different Sites

In the examples shown in Sections 3.1 and 3.2, optimizing the turbines layouts and yaw angles concurrently with our geometric yaw relationship resulted in a 6.5% and a 0.8% improvement in the design objective, respectively, compared to the sequential method of optimizing layout then yaw angles. These examples are simplified for this paper. For example, the scenario shown in Section 3.2 with the spatially varying inflow has a dense wind plant (20 MW/km$^2$) and a minimum spacing constraint of two turbine rotor diameters, where real wind plants are often much less dense and have larger minimum spacing constraints. While a full understanding of the gains possible through coupled layout and yaw angle optimization would require work well beyond the scope of this paper, we can provide intuition about which conditions could benefit the most from a control co-design approach.

Wind plants that would benefit the most from control co-design share one over-arching trait; they have the competing priorities of trying to minimize wake interference while also decreasing the spacing between the turbines. These competing priorities could arise from spatially varying wind resource, spatially varying costs (e.g., soil conditions, water depth for offshore turbines), objectives benefiting from tightly packed turbines (e.g., relatively expensive array cables, maximizing capacity while maintaining some desired efficiency), etc. Situations that would not benefit as much from control co-design are those that would trend towards similar layouts regardless of if the layout optimization is performed with or without coupled yaw control. These attributes are those opposite of the previously listed. Additionally, this could include wind plants that are highly constrained spatially, such that there is not much freedom in the layout optimization. In this case, the layout-and-yaw coupled solution would be similar to or the same as the sequential one.

In terms of the magnitude of potential upside available from layout and yaw control co-design, we expect this to be highly variable from site to site. We expect that the 0.8% gain reported in Section 3.2 is on the high end of potential benefits because

of the high power density and small minimum spacing constraint. However, we do expect non-negligible improvements on the same order as our example for many real wind plants, namely those with many or all of the traits listed above, as those likely to benefit greatly from layout and yaw control co-design.

## 4 Conclusions

In this article, we presented a geometric yaw relationship that can be used to determine sufficiently optimal wind turbine yaw angles for wake steering as a function of the layout of wind power plants. This method, or any improvement on the specific relationship presented in this article, can be used to solve the coupled wind plant layout and yaw control optimization problem in a computationally efficient manner, and can find layouts that perform significantly better than layouts that are optimized without yaw. In Section 3.2, we describe how we used geometric yaw to obtain a plant layout that performed 0.8% better than a layout optimized assuming no yaw, with no difference in the number of function calls or computation time required to optimize. At the scale wind farms are being designed and to meet ambitious renewable energy goals worldwide, 0.8% is a significant improvement in plant performance. We expect that many wind plants of the future will benefit similarly through the control co-design approach enabled by our approach. The geometric yaw relationship presented in this article enables fully coupled wind plant layout and yaw control optimization with no added expense compared to the regular wind plant layout optimization problem, and can greatly improve how wind plant layout optimization is approached by researchers and wind power plant developers alike.

*Code availability.* https://github.com/pjstanle/GeometricYaw/tree/paper/initial_submission

*Author contributions.* AS was responsible for Conceptualization, Investigation, Methodology, Software, Visualization, and Writing – original draft preparation. CB was responsible for Conceptualization, Investigation, Methodology, Software, and Writing – review and editing. PF was responsible for Funding acquisition, Project administration, Resources, Supervision, and Writing – review and editing.

*Competing interests.* Paul Fleming is a member of the editorial board of *Wind Energy Science*.

*Acknowledgements.* This work was authored by the National Renewable Energy Laboratory, operated by Alliance for Sustainable Energy, LLC, for the U.S. Department of Energy (DOE) under Contract No. DE-AC36-08GO28308. Funding provided by the U.S. Department of Energy Office of Energy Efficiency and Renewable Energy Wind Energy Technologies Office. The views expressed in the article do not necessarily represent the views of the DOE or the U.S. Government. The U.S. Government retains and the publisher, by accepting the

article for publication, acknowledges that the U.S. Government retains a nonexclusive, paid-up, irrevocable, worldwide license to publish or reproduce the published form of this work, or allow others to do so, for U.S. Government purposes.

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
