# Peer review of "Enabling Control Co-Design of the Next Generation of Wind Power Plants"

_Wind Energy Science, 2023_

## Referee Comment (RC1)

**Generel comments:**

- Summary: Layout design and wake steering (achieved through yaw control) is combined in a co-design process that does not exceed the computational cost of a traditional layout optimization by determining yaw angles from the layout.
- The work is well-placed in context to related literature and clearly motivated.
- The reported 0.8% increase in AEP is indeed a very interesting result – but how generally applicable is it? In this paper it is based on one comparison. It would be interesting to have further comments on this and the limitations of their work from the authors.
- All in all a useful, very well-written and interesting study.

**Specific comments:**

Section 1. Introduction
- p. 3. Fig. 1: As a reader, it would be helpful with additional information in order to better relate to the stated computation time for an optimization. What are the computational resources used (e.g., amount and type of CPUs)? How well are these optimization problems converged (in SNOPT that would be the Major Optimality Tolerance)? How are the gradients computed (further down on p. 9 finite difference is mentioned but that is for the SciPy setup)? The information cannot fit into a single figure caption so one idea could also be to fit them into the Section 2 on the Geometric Yaw Relationship where SNOPT is mentioned. Still, without more information it is difficult for a reader to relate to the reported computation time.
- p. 3. Fig. 1: Why are there three dots on the curve?
- p. 3: 'Revolutionize' is as a word brought up three times in this paper. Abstract, here, and finally in the Conclusions (where it says 'can revolutionize' and not will revolutionize). Indeed, the method is clearly an improvement – but the 0.8% presented further down is based on one comparison. One could perhaps modify the wording slightly here – or to build confidence of the significance for the findings presented in this study - somewhere include comments of how generally applicable these results are. What about some of the factors not taking into considerations (comparing orders of magnitude in effect)?

Section 2. Geometric Yaw Relationship
- p. 4: See comment on Fig 1 above. It would be interesting to have more information on the optimizations.
- p. 4 On the geometric yaw relationship:
  It is based on the (very illustrative) plot in Fig. 2C. Still, the resulting yaw relationship in Fig 2.D seems 'manually' constructed (i.e., "defined through observation of the pattern"). Why not simply interpolate using e.g., a piecewise linear model or perhaps a curvature-minimizing interpolant in 2D? Could the authors' reasoning be, that these potential improvements would only lead to minimal extra gain? This could be further elaborated.

Section 3: Results
- p. 6: suggestion: SciPy instead of scipy.
- p. 6: Spelling typo: SLSQP gradient-based optim[i]zer
- p. 6: What was the reason for changing optimization setup from SNOPT to SciPy?
- p. 6: Was it necessary to add the final continuous yaw optimization in the second example where the formulation leveraging geometric yaw relationship is tested? It would seem, that the ideal incorporation of the geometric yaw in the optimization procedure would not need a final yaw optimization at the very end. How much improvement is gained from this extra optimization? Could this extra step be avoided somehow (e.g., through a more accurate

geometric yaw relationship based on interpolation)? More comments on this would be interesting.

- p. 9: Spelling typo: 0.8 % higher AEP [that] → 0.8% higher AEP [than].
- p. 9: When comparing the final layout result for the two optimization strategies (seen in Fig. 4C-D) it would be interesting to also see the starting position of the two layouts. This is relevant since the final part of the Results section speaks to the disparate final layouts obtained with the two methods. This is indeed a clear tendency also seen from the illustrative 1-D example. Still, a contributing factor not mentioned could also be, that the two strategies started from different baseline layouts (each method could start from any of the 50 candidate layouts). This could be further clarified in the text.
- p. 9: Further information on the two final optimizations being compared would be of interest. How many function calls for each optimization? How well were the optimization problems converged? Comments on the computational resources, computation time used, etc.

Section 4: Conclusions
- p. 9: New information concerning number of function calls is introduced for the first time in this section. Ideally, this point should be introduced and clarified further already in the Results section before being mentioned in the conclusion. Other than that it is a nice, succinct conclusion.

---

## Author Comment (AC1)

**Response to Adam Stock**

Andrew P. J. Stanley, Christopher Bay, and Paul Fleming

May 2023

Adam, thank you for taking the time to read and provide feedback on this paper. We are very excited about this work, and we greatly appreciate your help to make this work as impactful as we think it can and should be.

We will incorporate your feedback into our revised paper. Below is a summary of your points in blue, and our response and the changes we will make in the revised paper immediately following shown in black.
* * *
It would be interesting to hear the author's views on the real-world applicability of the methodology. The control codesign methodology appears to only consider the energy capture of the turbines and does not consider other implications of a layout design such as accessibility, water depth or cabling lengths and layouts. Whilst one would not expect these aspects to be fully costed in and included in the analysis, the absence of these factors should be noted in the text with perhaps a comparison in the order of magnitude difference that such aspects may impart. Would these factors be easy to accommodate within the methodology? Is this a topic for future work perhaps?

This is an excellent point and one that we will address in the revised manuscript. Wind farm design is a very large problem involving many disciplines. In the paper we address layout optimization for yield, and one aspect of control. In a full wind farm one also needs to consider turbine selection or design, costs from foundations and cabling, accessibility for maintenance, impact to local species, social issues and acceptance, permitting... and the list goes on. Because of the wide variety of disciplines, and complex, computationally expensive processes of each of these aspects of design many of these processes are separated, with the design performed sequentially (i.e., first layout optimization for yield, then foundations, then electrical...). The ideal would be to eventually couple **all** aspects of design into a single process such that tradeoffs and sensitivities are fully captured enabling better wind farm design. There are several reasons why this is not currently done, a few being computational expense, legacy processes (depending on the institution), and different fidelities of models.

In this paper we provide a solution to one aspect of wind farm codesign, specifically layout optimization plus wake steering optimization. In the revised manuscript we will make this clear, and clearly state that full wind farm design is a more complex process to which we aspire to couple into a fully coupled multidisciplinary optimization process.
* * *
The text notes that "we expect larger comparative gains for larger plants with more turbines" but the basis for this assertion is not presented. Is it not the case that, beyond a certain point, wakes can become somewhat of a "wake soup" and there could be diminishing returns? Some discussion of the implications of applying the method on farms with more turbines would be useful.

Thanks for this feedback. We will address this in the revised manuscript to provide justification and/or further examples with more turbines.
* * *
The equation for the wake expansion r_wake = 0.1x + r_turbine seems broadly sensible, but there is no reference for this equation. Might this value change in different atmospheric conditions? Does the

wake continue expanding indefinitely? Furthermore, it would be useful for the authors to comment on the sensitivity of their results to changes in the wake expansion factor.

Yes this is a good point, as the implicit yaw angle relationship will certainly change depending on how wide a wake is assumed to be. The wake expansion is taken from the original formulation of the Jensen wake model. We will comment on this in the revised manuscript, and add discussion about how this could affect the implicit relationship, and how it could be improved in future iterations.
* * *
Alongside the result of "0.8% more energy than its sequentially designed counterpart" it would be informative to know the improvement compared to purely optimising layout with no WFFC applied.

Agreed. This is important information that we will include in the revision.
* * *
Why was the particular wind rose that was used chosen? Similar to the wake expansion point above, what sort of sensitivity does the method have to different wind roses? As a wider point, the work presents a single example – can the authors be sure that the result is typical? What are the bounds of that "typicality"?

Excellent point, this is a shortcoming in the preprint. We will add discussion about what we expect the sensitivity of the results depending on the many inputs to be.

---

## Author Comment (AC2)

**Response to RC1**

Andrew P. J. Stanley, Christopher Bay, and Paul Fleming

May 2023

Before we proceed with our response to your comments, we want to express sincere gratitude for your time and energy spent acting as a reviewer for this paper. We recognize that you have taken time out of a busy schedule to help our work to be more clear and impactful, and we appreciate your contribution.

We have incorporated your feedback into our revised paper. Below is a summary of your points in blue, and our response and changes in the paper immediately following shown in black.

**1 General comments**

Summary: Layout design and wake steering (achieved through yaw control) is combined in a co-design process that does not exceed the computational cost of a traditional layout optimization by determining yaw angles from the layout. The work is well-placed in context to related literature and clearly motivated.
* * *
The reported 0.8% increase in AEP is indeed a very interesting result – but how generally applicable is it? In this paper it is based on one comparison. It would be interesting to have further comments on this and the limitations of their work from the authors.

Thank you for this feedback, it is something we fully agree with and think further discussion on this matter will be a significant improvement on the preprint version. In the revised paper, we will add discussion about our example layout optimization results. Specifically, we will discuss the potential sensitivity of the results to the various problem parameters and in what situations we expect this method to perform particularly well, and where it might be less impactful.
* * *
All in all a useful, very well-written and interesting study.

Thank you! We are very excited about this work, and the implications it may have to other aspects of wind farm codesign.

**2 Specific comments**

**2.1 Section 1. Introduction**

p. 3. Fig. 1: As a reader, it would be helpful with additional information in order to better relate to the stated computation time for an optimization. What are the computational resources used (e.g., amount and type of CPUs)? How well are these optimization problems converged (in SNOPT that would be the Major Optimality Tolerance)? How are the gradients computed (further down on p. 9 finite difference is mentioned but that is for the SciPy setup)? The information cannot fit into a single figure caption so one idea could also be to fit them into the Section 2 on the Geometric Yaw Relationship where SNOPT is mentioned. Still, without more information it is difficult for a reader to relate to the reported computation time.

We agree this is important information to include. In the revised manuscript, we will add discussion about the computational resources used for the results in the paper, as well as information about the problem setup.

The three dots represent the three fully coupled layout and yaw optimizations that we ran to understand this computational expense relationship. We will clarify this in the revised paper.

p. 3: 'Revolutionize' is as a word brought up three times in this paper. Abstract, here, and finally in the Conclusions (where it says 'can revolutionize' and not will revolutionize). Indeed, the method is clearly an improvement – but the 0.8% presented further down is based on one comparison. One could perhaps modify the wording slightly here – or to build confidence of the significance for the findings presented in this study - somewhere include comments of how generally applicable these results are. What about some of the factors not taking into considerations (comparing orders of magnitude in effect)?

This is a great point. We were bold with this language, perhaps without providing sufficient evidence to justify it. In the revised draft we will incorporate your feedback, to modify wording or better build confidence of the significance of these findings.

**2.2 Section 2. Geometric Yaw Relationship**

p. 4: See comment on Fig 1 above. It would be interesting to have more information on the optimizations.

Agreed, in the revision we will provide more detailed information.

p. 4 On the geometric yaw relationship: It is based on the (very illustrative) plot in Fig. 2C. Still, the resulting yaw relationship in Fig 2.D seems 'manually' constructed (i.e., "defined through observation of the pattern"). Why not simply interpolate using e.g., a piecewise linear model or perhaps a curvature-minimizing interpolant in 2D? Could the authors' reasoning be, that these potential improvements would only lead to minimal extra gain? This could be further elaborated.

We agree, this needs to be more clear. You are correct that the relationship used for the results in this paper was manually constructed. This was intentional to illustrate that significant improvements can be achieved with such a simple relationship. This will be clarified in the revision. Additionally we will add discussion that this is a specific area for improvement as we develop this method further.

**2.3 Section 3: Results**

This will be corrected in the revised manuscript.

p. 6: Spelling typo: SLSQP gradient-based optim[i]zer

This will be corrected in the revised manuscript.

p. 6: What was the reason for changing optimization setup from SNOPT to SciPy?

This change in optimizer was purely a logistical one. The lead author who ran the optimizations changed organizations shortly before running them, and did not have access to SNOPT in the new organization. We don't think this is necessary information to include in the revision, although we are happy to add some explanation if you think it is important.

p. 6: Was it necessary to add the final continuous yaw optimization in the second example where the formulation leveraging geometric yaw relationship is tested? It would seem, that the ideal incorporation of the geometric yaw in the optimization procedure would not need a final yaw optimization at the very end. How much improvement is gained from this extra optimization? Could this extra step be avoided somehow (e.g., through a more accurate geometric yaw relationship based on interpolation)? More comments on this would be interesting.

This is a great comment. In short, yes the final continuous yaw optimization is necessary. The geometric yaw relationship underperforms the continuous yaw optimization fairly significantly in the current form. This information in itself is quite interesting however. The geometric yaw relationship does not need to perform very closely to a fully continuous optimization! It just needs to perform well enough that when used in a codesign framework, wake steering is sufficiently accounted for during the layout optimization and a better solution is achieved. In the revision, we will add discussion on this matter, and also clarify that an improved geometric yaw relationship may enable us to find even better solutions, and remove the requirement of a final continuous yaw optimization step.
* * *
p. 9: Spelling typo: 0.8% higher AEP [that] to 0.8% higher AEP [than].

This will be corrected in the revised manuscript.
* * *
p. 9: When comparing the final layout result for the two optimization strategies (seen in Fig. 4C-D) it would be interesting to also see the starting position of the two layouts. This is relevant since the final part of the Results section speaks to the disparate final layouts obtained with the two methods. This is indeed a clear tendency also seen from the illustrative 1-D example. Still, a contributing factor not mentioned could also be, that the two strategies started from different baseline layouts (each method could start from any of the 50 candidate layouts). This could be further clarified in the text.

Thank you for this comment, we agree this is important to clarify and discuss better. The 50 starting layouts were generated in a preprocessing step. Then, each of the 50 baseline layouts was optimized with both methods (with and without the geometric yaw relationship). In short, each method had the same set of starting conditions. Regardless, this is critical to explain better, and we will clarify this in the revision.
* * *
p. 9: Further information on the two final optimizations being compared would be of interest. How many function calls for each optimization? How well were the optimization problems converged? Comments on the computational resources, computation time used, etc.

Thank you for this comment, we will add discussion about the computational resources, computational expense, convergence, and other information about the optimizations in the revision.

**2.4   Section 4: Conclusions**

p. 9: New information concerning number of function calls is introduced for the first time in this section. Ideally, this point should be introduced and clarified further already in the Results section before being mentioned in the conclusion. Other than that it is a nice, succinct conclusion.

Thank you! And yes, as mentioned in the previous comment we will add discussion about computational expense in the results section.

---

## Author Comment (AC3)

**Response to Sebastian Sanchez Perez-Moreno**

Andrew P. J. Stanley, Christopher Bay, and Paul Fleming

May 2023

Before we begin our response, we want to sincerely express our gratitude. Sebastian, as an expert in this field and busy researcher we know that your time is valuable. We truly are grateful that you have taken your time to review this paper and provide excellent feedback so that this paper can reach the potential that we think it is capable.

We have incorporated your feedback into our revised paper. Below is a summary of your points in blue, and our response and changes in the paper immediately following shown in black

**1  General comments**

This paper provides a novel approach to co-optimise the wind turbine layout and yaw angles for wake steering. This method greatly accelerates the co-design process by avoiding nested optimisation loops that are computationally very expensive. Together with the understanding of the dependency of the optimal yaw angle with respect to the relative position of the nearest waked wind turbine, these are the biggest contributions from this research. The outcome of this paper is directly applicable in engineering processes in our industry.

It is enourmously appreciated the transparency and repeatability that comes with the code being publicly available. Also want to congratulate the authors on the high quality of the figures, which communicate the outcomes more clearly.

Thank you, we are very excited about this work and how similar approaches could be used for other aspects of codesign.
* * *
I would like to read follow up research ideas stemming from this first step, to reduce the uncertainty and risks associated with the assumptions made here (power density, wake model). With more research, industry would deploy this method with more confidence, as the layouts optimised with co-design perform worse than without it, when not operated with wake steering.

We fully agree. There are several opportunities for continued research. We see the primary opportunities as 1) improving the geometric yaw relationship including better quasi-optimal yaw angles, and exploring the validity for different modeling parameters, 2) improving the objective function to capture more complex operation (a wind farm will neither operate fully with wake steering nor fully without wake steering), and 3) expanding a similar approach to other aspects of codesign where we could develop an implicit relationship between layout and some other variable. We will improve this discussion about future opportunities in the revised paper.
* * *
Additionally, could the authors mention something with respect to the turbine fatigue loads expected from implementing these yaw angles?

Yes, thank you for this comment. This is lacking in the preprint paper, and we will add some discussion in the revision on the impact of yaw offset on fatigue loads.

**2 Specific comments**

Line 2: it says "...currently optimized separately... more and more wind plants implement wake steering as their primary form of operation" (gives impression that wake steering is already implemented in wind plants in operation, is this true?)

Great point. We will reword this in the revised manuscript.
* * *
Line 54: "...per wind condition". What is meant here, wind speed and direction? Can you be more specific?

This will be changed to "per wind speed and wind direction combination" in the revised manuscript.
* * *
Figure 1: Don't the number of wind speed bins count? Or is a single wind speed per direction used to make this figure? This could be stated explicitly. Over 150 hours for a "small" wind farm is already too long for a single wind speed.

Good point. We will clarify this in the revision. For this figure, a single speed bin per wind direction was used (which yes is already far too long).
* * *
Line 75: In my opinion, it's somewhat exaggerated the statement that this new approach alone can accelerate the deployment of wind energy and reach goals.

We agree, this is probably overstated. We will reword in the revised manuscript.
* * *
Line 85: All downstream turbines are waked (if by small amounts) according to Gaussian models, so it would good to be explicit say what model is used here (looks like a top hat Jensen profile). And can you mention what wake decay factor corresponds roughly to the wake radius formula? I am curious about how "optimistic" this wake radius is and how the conclusions can change depending on this wake expansion factor.

This is a good point, and one that we will clarify in a revised version. You are correct that all downstream turbines would be waked if using a Gaussian wake model. Therefore, for this paper we used two separate wake calculations.

1. Calculate which downstream turbines are waked. You can imagine several ways to do this. If you wanted to use a Gaussian model, you could assume the wake width is everywhere the wake deficit is greater than X%. For this paper, you are correct that we used a Jensen top hat model to determine if downstream turbines were waked. The important thing for the purposes of our geometric yaw relationship is that this is a Boolean value; a turbine is either waked or not.

2. Perform a full wake calculation for the wind farm energy production. This model can be different than the one used to determine "waked-ness," and is in this paper.

Hopefully that is clear, certainly reach out if you have further questions. Regardless, in the revised paper we will add discussion on this topic, including how the results and model may change based on different assumptions.
* * *
Fig 2c: are there orange or blue coloured points underneath the black dots? Are there clearly many more black dots than coloured? How did you determine the 1D threshold?

These are good questions. We will revisit the figure generation to make sure all of the points are appropriately represented and many points aren't hiding behind others. If we understand the last question correctly, the 1D threshold was determined manually by inspecting the figure 2c. This is clearly an opportunity for improvement in future research.
* * *
Line 121: why do you re-optimise the yaw angle after a layout has been optimised with co-design? What are the quantitative improvements before this step?

This is a good question and is similar to a question from the other reviewer. We will copy part of our response here:

"In short, yes the final continuous yaw optimization is necessary. The geometric yaw relationship underperforms the continuous yaw optimization fairly significantly in the current form. This information in itself is quite interesting however. The geometric yaw relationship does not need to perform very closely to a fully continuous optimization! It just needs to perform well enough that when used in a codesign framework, wake steering is sufficiently accounted for during the layout optimization and a better solution is achieved. In the revision, we will add discussion on this matter, and also clarify that an improved geometric yaw relationship may enable us to find even better solutions, and remove the requirement of a final continuous yaw optimization step."
* * *
Do the final re-optimised yaw values correlate nicely with the deterministic yaw angles found during the co-optimisation?

Good question, this relates to the previous question as well. In short, no not really. They are different enough that before performing the continuous yaw optimization, the codesign layout underperforms the sequential one (but, that is not really a 1 to 1 comparison). This is an area for improvement in further development of this geometric yaw relationship. In the revised paper, we will discuss and clarify this last continuous yaw optimization step to clarify all of these elements.
* * *
Line 115: doesn't SLSQP require multiple initial conditions to get closer the global optimum? What were the initial conditions (layout) in 3.1?

Good questions. As discussed in the paper, for 3.2 we used 50 different starting layouts. For 3.1 we assumed the design space was simple enough that we only used one starting set of design variables (all turbines spaced 5 rotor diameters apart). In the revised version we will repeat this optimization for multiple starting layouts for example 3.1, and add the appropriate discussion on the topic.
* * *
What is the turbine nameplate capacity in example 3.2? What is the plant power density? 2 km x 2 km seems small even for a 3.5 MW WTG rated capacity (14 MW/km2). Is the AEP increase of co-design as big as 0.8% for sites with smaller but more realistic power density (e.g. 5 MW/km2)?

Excellent questions, these are important to clarify. In the revised paper we will add explanation of the wind plant characteristics, as well as discussion on how the 0.8% figure may apply to other sites.
* * *
Does wake steering optimisation for existing plants still have any value after finding this geometric relationship? What's the trade-off between "slow" optimisation and finding the optimal angles deterministically?

These are good questions. There is value to have a deterministic relationship between turbine locations and optimal yaw angles. Because turbines are sometime down for routine or unscheduled maintenance, to is infeasible to have a look up table for every possible combination of turbines that may be operational at a given time. One possible solution to this issue is to improve this geometric yaw relationship (or something similar) to better approach the continuous yaw optimization solution. We will add discussion of this in the revised manuscript.

**3 Technical corrections**

Figure 1: shouldn't it say "with 24 wind direction bins" instead of "with the 24 wind direction bins"?

This will be changed to "using 24 wind direction bin" in the revised manuscript.

Line 115: typo - should say "optimizer" where it says "optimzer"

This will be corrected in the revised manuscript.

---

## Author Response (AR1)

**Response to Reviewers**

Andrew P. J. Stanley, Christopher Bay, and Paul Fleming

June 2023

**Response to Reviewer 1**

Before we proceed with our response to your comments, we want to express sincere gratitude for your time and energy spent acting as a reviewer for this paper. We recognize that you have taken time out of a busy schedule to help our work to be more clear and impactful, and we appreciate your contribution.

We have incorporated your feedback into our revised paper. Below is a summary of your points in blue, our response, and changes in the paper immediately following shown in black.

Please note that some of this text is the same as our original response to your comments. We have modified the same document to more specifically indicate the changes we made the the revised manuscript.

**1 General comments**

Summary: Layout design and wake steering (achieved through yaw control) is combined in a co-design process that does not exceed the computational cost of a traditional layout optimization by determining yaw angles from the layout. The work is well-placed in context to related literature and clearly motivated.
* * *
The reported 0.8% increase in AEP is indeed a very interesting result – but how generally applicable is it? In this paper it is based on one comparison. It would be interesting to have further comments on this and the limitations of their work from the authors.

Thank you for this feedback, it is something we fully agree with and think further discussion on this matter will be a significant improvement on the preprint version.

In the revised draft, we have added a new section 3.3 discussing this matter, and reworded the conclusion appropriately.
* * *
All in all a useful, very well-written and interesting study.

Thank you! We are very excited about this work, and the implications it may have to other aspects of wind farm codesign.

**2 Specific comments**

**2.1 Section 1. Introduction**

p. 3. Fig. 1: As a reader, it would be helpful with additional information in order to better relate to the stated computation time for an optimization. What are the computational resources used (e.g., amount and type of CPUs)? How well are these optimization problems converged (in SNOPT that would be the Major Optimality Tolerance)? How are the gradients computed (further down on p. 9 finite difference is mentioned but that is for the SciPy setup)? The information cannot fit into a single figure caption so one idea could also be to fit them into the Section 2 on the Geometric Yaw Relationship where SNOPT is mentioned. Still, without more information it is difficult for a reader to relate to the reported computation time.

Thank you for this comment! Figure 1 is meant to be illustrative of the poor scaling of computational expense with increasing design variables only, for that reason we don't believe an exhaustive description and definition of the optimizations is necessary. However, we agree that more information is helpful for the reader to relate to the problem and understand it better. In the revised manuscript, we have added more description of the optimization in the text of Section 1 (the introduction), as well as clarified information about the convergence in the caption of Figure 1.
* * *
p. 3. Fig. 1: Why are there three dots on the curve?

The three dots represent the three fully coupled layout and yaw optimizations that we ran to understand this computational expense relationship.

In the revised paper, we have rewritten the caption to Figure 1 to make sure this is clearly explained.
* * *
p. 3: 'Revolutionize' is as a word brought up three times in this paper. Abstract, here, and finally in the Conclusions (where it says 'can revolutionize' and not will revolutionize). Indeed, the method is clearly an improvement – but the 0.8% presented further down is based on one comparison. One could perhaps modify the wording slightly here – or to build confidence of the significance for the findings presented in this study - somewhere include comments of how generally applicable these results are. What about some of the factors not taking into considerations (comparing orders of magnitude in effect)?

This is a great point. We were bold with this language, perhaps without providing sufficient evidence to justify it.

As mentioned previously, in the revised draft we have added a new section 3.3 discussing the generally applicability of the 0.8% that we reported, and reworked a small part of the conclusion. Additionally, we have removed or reworded our the work "revolutionize" to be more conservative in our claims.

**2.2   Section 2. Geometric Yaw Relationship**

p. 4: See comment on Fig 1 above. It would be interesting to have more information on the optimizations.

We fully agree. We have added the following to the revised manuscript, which includes additional details and reminds readers that they can also refer directly to the run scripts.

"For these yaw optimizations, we used finite-difference gradients, bounds between -30 and 30 degrees for turbine yaw angles, and default convergence settings. For additional information regarding these optimizations, please see the run scripts in the code referenced at the end of this paper."
* * *
p. 4 On the geometric yaw relationship: It is based on the (very illustrative) plot in Fig. 2C. Still, the resulting yaw relationship in Fig 2.D seems 'manually' constructed (i.e., "defined through observation of the pattern"). Why not simply interpolate using e.g., a piecewise linear model or perhaps a curvature-minimizing interpolant in 2D? Could the authors' reasoning be, that these potential improvements would only lead to minimal extra gain? This could be further elaborated.

We agree, this needs to be more clear. You are correct that the relationship used for the results in this paper was manually constructed. This was intentional to illustrate that significant improvements can be achieved with such a simple relationship. In the revised manuscript, we have added an additional paragraph to the end of section 2 discussing this point, and possible improvements we hope to see in the near future.

**2.3   Section 3: Results**

p. 6: suggestion: SciPy instead of scipy.

This has been corrected in the revised manuscript.
* * *
p. 6: Spelling typo: SLSQP gradient-based optim[i]zer

This was corrected in the revised manuscript.
* * *
p. 6: What was the reason for changing optimization setup from SNOPT to SciPy?

This change in optimizer was purely a logistical one. The lead author who ran the optimizations changed organizations shortly before running them, and did not have access to SNOPT in the new organization. Because we have clearly explained our optimizers for each part of this research (although they are different at some stages), we don't think this change is necessary information to explain.
* * *
p. 6: Was it necessary to add the final continuous yaw optimization in the second example where the formulation leveraging geometric yaw relationship is tested? It would seem, that the ideal incorporation of the geometric yaw in the optimization procedure would not need a final yaw optimization at the very end. How much improvement is gained from this extra optimization? Could this extra step be avoided somehow (e.g., through a more accurate geometric yaw relationship based on interpolation)? More comments on this would be interesting.

This is a great comment. In short, yes the final continuous yaw optimization is necessary. The geometric yaw relationship underperforms the continuous yaw optimization fairly significantly in the current form. This information in itself is quite interesting however. The geometric yaw relationship does not need to perform very closely to a fully continuous optimization! It just needs to perform well enough that when used in a co-design framework, wake steering is sufficiently accounted for during the layout optimization and a better solution is achieved. In the revision, we will add discussion on this matter, and also clarify that an improved geometric yaw relationship may enable us to find even better solutions, and remove the requirement of a final continuous yaw optimization step.

The paragraph we added at the end of section two addresses some of these issues. We have also added the following text immediately preceding section 3.1 to address this comment:

"With the geometric yaw relationship shown in Figure 2D, the purpose is to sufficiently account for yaw control during the layout optimization to affect the optimal turbine locations. Continuously optimizing the yaw angles for wake steering outperforms those predicted by the geometric yaw model, so at least with this specific relationship the final continuous yaw optimization was necessary after the layout was fixed. Perhaps an improved geometric yaw relationship could remove the necessity of this last yaw angle optimization."
* * *
p. 9: Spelling typo: 0.8% higher AEP [that] to 0.8% higher AEP [than].

This was corrected in the revised manuscript.
* * *
p. 9: When comparing the final layout result for the two optimization strategies (seen in Fig. 4C-D) it would be interesting to also see the starting position of the two layouts. This is relevant since the final part of the Results section speaks to the disparate final layouts obtained with the two methods. This is indeed a clear tendency also seen from the illustrative 1-D example. Still, a contributing factor not mentioned could also be, that the two strategies started from different baseline layouts (each method could start from any of the 50 candidate layouts). This could be further clarified in the text.

Thank you for this comment, we agree this is important to clarify and discuss better. The 50 starting layouts were generated in a preprocessing step. Then, each of the 50 baseline layouts was optimized with both methods (with and without the geometric yaw relationship). In the updated manuscript, this was clarified as follows:

"Because of the large amount of local minima known to exist in the wind plant layout optimization problem, and because gradient-based optimizers are known to converge to local minima without full exploration of the design space, we repeated each optimization 50 times with randomly initialized design variables. This process was as follows:

1. Randomly initialize a starting turbine layout.

2. Perform the layout optimization twice from the starting layout in step 1, once with no yaw control and once with yaw control using the geometric yaw relationship.

3. Repeat steps 1 and 2 for a total of 50 different starting layouts.

Optimizing the each of the starting layouts with and without yaw control removed the possibility that differences in the optimized layouts and performances were due to different starting conditions."
* * *
p. 9: Further information on the two final optimizations being compared would be of interest. How many function calls for each optimization? How well were the optimization problems converged? Comments on the computational resources, computation time used, etc.

Thank you for this comment, we agree this is important information to include and greatly adds to this paper!

In the revised manuscript, we have added an explanation of the optimizer options used (most of them were default) and how gradients were computed. We added a statement that all optimizations terminated successfully with optimiality tolerance achieved. We have also added two figures and appropriate discussion with the computational expense (function calls and wall time) required for the two examples given.

**2.4   Section 4: Conclusions**

p. 9: New information concerning number of function calls is introduced for the first time in this section. Ideally, this point should be introduced and clarified further already in the Results section before being mentioned in the conclusion. Other than that it is a nice, succinct conclusion.

Thank you! And yes, as mentioned in the previous comment we added discussion about computational expense in the results section.

**Response to Reviewer 2**

Before we begin our response, we want to sincerely express our gratitude. Sebastian, as an expert in this field and busy researcher we know that your time is valuable. We truly are grateful that you have taken your time to review this paper and provide excellent feedback so that this paper can reach the potential that we think it is capable.

We have incorporated your feedback into our revised paper. Below is a summary of your points in blue, our response, and changes in the paper immediately following shown in black

Please note that some of this text is the same as our original response to your comments. We have modified the same document to more specifically indicate the changes we made the the revised manuscript.

**1 General comments**

This paper provides a novel approach to co-optimise the wind turbine layout and yaw angles for wake steering. This method greatly accelerates the co-design process by avoiding nested optimisation loops that are computationally very expensive. Together with the understanding of the dependency of the optimal yaw angle with respect to the relative position of the nearest waked wind turbine, these are the biggest contributions from this research. The outcome of this paper is directly applicable in engineering processes in our industry.

It is enourmously appreciated the transparency and repeatability that comes with the code being publicly available. Also want to congratulate the authors on the high quality of the figures, which communicate the outcomes more clearly.

Thank you, we are very excited about this work and how similar approaches could be used for other aspects of codesign.
* * *
I would like to read follow up research ideas stemming from this first step, to reduce the uncertainty and risks associated with the assumptions made here (power density, wake model). With more research, industry would deploy this method with more confidence, as the layouts optimised with co-design perform worse than without it, when not operated with wake steering.

We fully agree. There are several opportunities for continued research. We see the primary opportunities as 1) improving the geometric yaw relationship including better quasi-optimal yaw angles, and exploring the validity for different modeling parameters, 2) improving the objective function to capture more complex operation (a wind farm will neither operate fully with wake steering nor fully without wake steering), and 3) expanding a similar approach to other aspects of co-design where we could develop an implicit relationship between layout and some other variable.

In the revised paper, we have added a paragraph a the end of Section 2 that addresses these issues, and introduces some ideas for future work.
* * *
Additionally, could the authors mention something with respect to the turbine fatigue loads expected from implementing these yaw angles?

Load impacts from wake steering is a big topic which we did not address in this paper as we concern ourselves with the power optimization problem. A recent study by Kanev et al (Kanev, S.; Bot, E.; Giles, J. Wind Farm Loads under Wake Redirection Control. Energies 2020, 13, 4088. https://doi.org/10.3390/en13164088) indicates that while the implementation of the yaw angles on a given turbine increases certain fatigue loads, this can be offset by reduction in wake loading on the same turbine as a result of other turbines wake steering. This nets out to an improvement in fatigue loads. Still another topic is wake steering optimization considering loading impacts (Navalkar et al. in ACC 2023, I don't see this paper online, but we have referenced it in the paper), but this is also outside the scope of this paper.

In the revised paper, we have added a paragraph at the end of Section 2 which discusses loading on a turbine due to wake steering, and how our model could account for this with future work.

**2   Specific comments**

Line 2: it says "...currently optimized separately... more and more wind plants implement wake steering as their primary form of operation" (gives impression that wake steering is already implemented in wind plants in operation, is this true?)

Great point, we agree.

The revised manuscript says: "They are currently optimized separately, but with more and more computational and experimental studies demonstrating the gains possible through wake steering, there is a growing need from industry and regulating bodies to combine the layout and control optimization in a co-design process."
* * *
We agree, this is unclear.

The revision says: "per wind speed and wind direction combination" in the revised manuscript.
* * *
Figure 1: Don't the number of wind speed bins count? Or is a single wind speed per direction used to make this figure? This could be stated explicitly. Over 150 hours for a "small" wind farm is already too long for a single wind speed.

We agree, this should be clarified.

In the body of the text, the revision now says: "This relationship means that the computational expense required to run the fully coupled optimization scales very poorly as the number of turbines increases, a challenge often called the "curse of dimensionality." For an average-sized wind plant ($\sim$ tens of turbines), the fully-coupled problem can easily reach thousands or tens of thousands of coupled design variables. Figure 1 shows the wall time required to run a fully coupled layout and yaw control optimization versus the number of wind turbines. This figure highlights two characteristics of this problem. First, the time to optimize scales non-linearly with the number of turbines. Second, even with the small wind farms optimized in the creation of this figure, the wall time for the fully coupled problem is far too long for most applications. While the absolute value of this metric could be reduced through advanced computing capabilities and finely tuned optimizer settings, the principle remains that the fully coupled optimization problem is computationally expense, especially for large wind farms."

The Figure 1 caption now says: "The time to solve the coupled wind turbine layout and yaw control optimization problem as a function of the number of wind turbines. These optimizations were performed with the gradient-based Sparse Nonlinear OPTimizer (SNOPT), using 24 wind direction bins and one wind speed bin for each wind direction. The three points represent the fully couple optimizations that we ran (with 4, 8, and 16 wind turbines), while the line shows an exponential fit to the three points."
* * *
We agree, this is probably overstated. We have removed this statement in the revised manuscript.
* * *
how "optimistic" this wake radius is and how the conclusions can change depending on this wake expansion factor.

This is a good point, and one that we will clarify in a revised version. You are correct that all downstream turbines would be waked if using a Gaussian wake model. Therefore, for this paper we used two separate wake calculations.

1. Calculate which downstream turbines are waked. You can imagine several ways to do this. If you wanted to use a Gaussian model, you could assume the wake width is everywhere the wake deficit is greater than X%. For this paper, you are correct that we used a Jensen top hat model to determine if downstream turbines were waked (which we have referenced in the revised draft). The important thing for the purposes of our geometric yaw relationship is that this is a Boolean value; a turbine is either waked or not.

2. Perform a full wake calculation for the wind farm energy production. This model can be different than the one used to determine "waked-ness," and is in this paper.

Hopefully that is clear, certainly reach out if you have further questions.

In the revised paper, we have added a citation to the wake radius definition that we used in the geometric yaw relationship to determine if a turbine is waked or not. We have also added information in the results section that we used the Gauss-Curl-Hybrid model in FLORIS to evaluate plant performance.
* * *
Fig 2c: are there orange or blue coloured points underneath the black dots? Are there clearly many more black dots than coloured? How did you determine the 1D threshold?

These are good questions. If we understand the last question correctly, the 1D threshold was determined manually by inspecting the figure 2c. This is clearly an opportunity for improvement in future research.

In response to this comment, we have added a paragraph to the end of section 2 discussing some of the improvements that could be made to the model in the future, and more explicitly stating that the geometric yaw relationship in Figure 2D is derived manually. We have also increased the size of the points in Figure 2C to make it more clear that there are a few exceptions to the "1D offset" threshold we use for the geometric relationship.

In response to your question about whether certain data points are obscured by one plotted on top of them, yes that is undoubtedly the case. This figure is attempting to share over 100,000 data points, which can not fit easily into a small figure like this one and has resulted in some colored points obscured by black and some black points obscured by colored ones. It seems like the options available are to plot the color density instead of individual points, which would not cover individual points but would obscure the fact that this figure represents a very large number of individual yaw angles. The other option is to keep the original figure (with the point sizes increased as mentioned), which clearly demonstrates the large number of points but does result in some being covered. We think the latter is the appropriate choice for this figure. Although some individual points are covered, there are enough points such that statistically the correct trend is shown.
* * *
Line 121: why do you re-optimise the yaw angle after a layout has been optimised with co-design? What are the quantitative improvements before this step?

This is a good question and is similar to a question from the other reviewer. The following is the same response we provided to reviewer 1.

In short, yes the final continuous yaw optimization is necessary. The geometric yaw relationship under-performs the continuous yaw optimization fairly significantly in the current form. This information in itself is quite interesting however. The geometric yaw relationship does not need to perform very closely to a fully continuous optimization! It just needs to perform well enough that when used in a co-design framework,

wake steering is sufficiently accounted for during the layout optimization and a better solution is achieved. In the revision, we will add discussion on this matter, and also clarify that an improved geometric yaw relationship may enable us to find even better solutions, and remove the requirement of a final continuous yaw optimization step.

The paragraph we added at the end of section two addresses some of these issues. We have also added the following text immediately preceding section 3.1 to address this comment:

"With the geometric yaw relationship shown in Figure 2D, the purpose is to sufficiently account for yaw control during the layout optimization to affect the optimal turbine locations. Continuously optimizing the yaw angles for wake steering outperforms those predicted by the geometric yaw model, so at least with this specific relationship the final continuous yaw optimization was necessary after the layout was fixed. Perhaps an improved geometric yaw relationship could remove the necessity of this last yaw angle optimization."
* * *
Do the final re-optimised yaw values correlate nicely with the deterministic yaw angles found during the co-optimisation?

Good question, this relates to the previous question as well. In short, no not really. They are different enough that before performing the continuous yaw optimization, the co-design layout underperforms the sequential one (but, that is not really a 1 to 1 comparison). This is an area for improvement in further development of this geometric yaw relationship. The changes/additions to the paper noted in the previous comment address your comment here as well.
* * *
Line 115: doesn't SLSQP require multiple initial conditions to get closer the global optimum? What were the initial conditions (layout) in 3.1?

Good questions. As discussed in the paper, for 3.2 we used 50 different starting layouts. For 3.1 we initially assumed the design space was simple enough that we only used one starting set of design variables (all turbines spaced 5 rotor diameters apart). In the revised version we slightly modified the example optimization in section 3.1 such that the inter-turbine spacing was constant and represented by one design variable. We also repeated the optimization 50 times with a randomly initialized initial spacing. The explanation in section 3.1 was updated to explain these changes.
* * *
What is the turbine nameplate capacity in example 3.2? What is the plant power density? 2 km x 2 km seems small even for a 3.5 MW WTG rated capacity (14 MW/km2). Is the AEP increase of co-design as big as 0.8% for sites with smaller but more realistic power density (e.g. 5 MW/km2)?

Excellent questions, these are important to clarify.
In the revised paper we have added an additional section 3.3 which discusses some of these issues, where we expect control co-design to be most impactful, as well as discussion on how the 0.8% figure may apply to other sites.
* * *
Does wake steering optimisation for existing plants still have any value after finding this geometric relationship? What's the trade-off between "slow" optimisation and finding the optimal angles deterministically?

These are good questions. As mentioned in the paper, we believe that the primary benefit of our geometric yaw relationship is to be able to account for wake steering during wind farm layout optimization. However, with improvements to the model there could also be benefits for operating wind plants. Because turbines are sometimes down for routine or unscheduled maintenance, it is infeasible to have a look up table for the optimal yaw angle of every possible combination of turbines that may be operational at a given time. One possible solution to this issue is to improve our geometric yaw relationship (or something similar) to better approach the continuous yaw optimization solution. This would allow wind plant operators to immediately know turbine yaw angles for any combination of turbines that they have operational at a given time. Although this is an interesting topic, we believe that it is beyond the scope of this paper and have thus not made any additions to the paper based on this comment.

**3    Technical corrections**

Figure 1: shouldn't it say "with 24 wind direction bins" instead of "with the 24 wind direction bins"?

This was changed to "using 24 wind direction bin" in the revised manuscript.
* * *
Line 115: typo - should say "optimizer" where it says "optimzer"

This is corrected in the revised manuscript.

**Response to Community Reviewer**

Adam, thank you for taking the time to read and provide feedback on this paper. We are very excited about this work, and we greatly appreciate your help to make this work as impactful as we think it can and should be.

We have incorporated your feedback into our revised paper. Below is a summary of your points in blue, our response, and the changes we made in the revised paper immediately following shown in black.

Please note that some of this text is the same as our original response to your comments. We have modified the same document to more specifically indicate the changes we made the the revised manuscript.
* * *
It would be interesting to hear the author's views on the real-world applicability of the methodology. The control codesign methodology appears to only consider the energy capture of the turbines and does not consider other implications of a layout design such as accessibility, water depth or cabling lengths and layouts. Whilst one would not expect these aspects to be fully costed in and included in the analysis, the absence of these factors should be noted in the text with perhaps a comparison in the order of magnitude difference that such aspects may impart. Would these factors be easy to accommodate within the methodology? Is this a topic for future work perhaps?

This is an excellent point. Wind farm design is a very large problem involving many disciplines. In the paper we address layout optimization for yield, and one aspect of control. In a full wind farm one also needs to consider turbine selection or design, costs from foundations and cabling, accessibility for maintenance, impact to local species, social issues and acceptance, permitting... and the list goes on. Because of the wide variety of disciplines, and complex, computationally expensive processes of each of these aspects of design many of these processes are separated, with the design performed sequentially (i.e., first layout optimization for yield, then foundations, then electrical...). The ideal would be to eventually couple **all** aspects of design into a single process such that tradeoffs and sensitivities are fully captured enabling better wind farm design. There are several reasons why this is not currently done, a few being computational expense, legacy processes (depending on the institution), and different fidelities of models.

In this paper we provide a solution to one aspect of wind farm co-design, specifically layout optimization plus wake steering optimization. In the revised manuscript we have added two paragraphs to the end of Section 2 discussing improvements that could be made to the geometric yaw relationship, and the possible adjustments that could be made for optimization objectives beyond simply maximizing power/energy (specifically, also considering structural loads during the layout and control optimization).

As authors, we expect that a similar approach can be used to enable other aspects of co-design. That would entail finding some deterministic relationship between turbine layout and another aspect of wind plant design that is of interest. Then, the additional aspect could be accounted for during the layout optimization. While believe this is certainly possible, and hope that something like this is achieved in future work, we believe that it is beyond the scope of this paper and have not added any additional discussion on this topic.
* * *
The text notes that "we expect larger comparative gains for larger plants with more turbines" but the basis for this assertion is not presented. Is it not the case that, beyond a certain point, wakes can become somewhat of a "wake soup" and there could be diminishing returns? Some discussion of the implications of applying the method on farms with more turbines would be useful.

Thanks for this feedback, it is a great point.

In the revised manuscript we have removed this statement. Additionally we have added an addition section 3.3 which discusses when we expect control co-design to be most impactful, and how we expect our numbers to compare to other sites.
* * *
The equation for the wake expansion r_wake = 0.1x + r_turbine seems broadly sensible, but there is no reference for this equation. Might this value change in different atmospheric conditions? Does the wake continue expanding indefinitely? Furthermore, it would be useful for the authors to comment on the sensitivity of their results to changes in the wake expansion factor.

Good question. We had originally responded with the following:

"Yes this is a good point, as the implicit yaw angle relationship will certainly change depending on how wide a wake is assumed to be." — this is actually not the case. The yaw angle relationship we defined in the paper is normalized by the rotor diameter, not the wake radius. Thus, this definition of wake radius just needs to be sufficiently large to capture the downstream turbines, but not so large that it will include turbines that are not interacting with the upstream turbines through the wake.

In response to the rest of this comment, yes this portion of the model assumes the wake expands indefinitely. This is typical in many analytic wake models, thus something we feel is not necessary to make explicit in the revision. The wake expansion is taken from the original formulation of the Jensen wake model, and we have added the appropriate reference in the revision.
* * *
Alongside the result of "0.8% more energy than its sequentially designed counterpart" it would be informative to know the improvement compared to purely optimising layout with no WFFC applied.

Agreed. This is interesting information. The general information we want the reader to understand is that the optimizer will find a solution for the model it is given. So, if you tell the optimizer to assume no wake steering, it will find a solution that operates well without wake steering. On the flip side, if you optimize the layout while assuming wake steering, it will find a layout that performs well when operating with wake steering, but will underperform when operating without wake steering. This is a general trait of any mathematical optimizer, it will exploit the model and objective function that it is given. In the updated manuscript, we have added better discussion of this trait in Section 3.1, and believe that discussion is appropriate to convey this message.
* * *
Why was the particular wind rose that was used chosen? Similar to the wake expansion point above, what sort of sensitivity does the method have to different wind roses? As a wider point, the work presents a single example – can the authors be sure that the result is typical? What are the bounds of that "typicality"?

Excellent point, this is a shortcoming in the preprint. In the revised draft we have added a new Section 3.3 discussing the generality of our results, and we think this section is a great addition.

In regards to the exact wind rose that was chosen, we wanted a wind rose that would cause appreciable waking from multiple directions. Additionally, we wanted the wind rose to be binned finely enough to demonstrate that the layout and yaw co-design could be accomplished with our geometric yaw relationship, something that would be extremely hard or impossible to do with traditional methods that would fully couple all of the design variables.